# Comparative Evaluation of the Chemical Components and Anti-Inflammatory Potential of Yellow- and Blue-Flowered *Meconopsis* Species: *M. integrifolia* and *M. betonicifolia*

**DOI:** 10.3390/metabo14100563

**Published:** 2024-10-20

**Authors:** Peizhao Cheng, Ruixi Gan, Cong Wang, Qian Xu, Kelsang Norbu, Feng Zhou, Sixin Kong, Zhuoma Jia, Dawa Jiabu, Xin Feng, Junsong Wang

**Affiliations:** 1Center of Molecular Metabolism, Nanjing University of Science and Technology, Nanjing 210094, China; cpz@njust.edu.cn (P.C.); virginia0216@163.com (R.G.); congwang95@njust.edu.cn (C.W.); zhiqidangran@gmail.com (Q.X.); 2Tibet Ganlu Tibetan Medicine Co., Ltd., Lhasa 851400, China; 15708010815@139.com; 3Tibet Ganlu Pharmaceutical Technology Co., Ltd., Lhasa 851400, China; 18389016465@163.com (F.Z.); 17797631513@163.com (Z.J.); 15771985220@163.com (D.J.); 4Shiningherb (Beijing) International Bio-Tech Co., Ltd., Beijing 100073, China; shiningherb@sina.com; 5Tibetan Medicine Institute, China Tibetan Research Center, Beijing 100101, China

**Keywords:** *Meconopsis*, *Meconopsis integrifolia*, *Meconopsis betonicifolia*, anti-inflammation, metabolomics, traditional Tibetan medicine, chemical composition

## Abstract

**Background/Objectives:** *Meconopsis* has long been used in traditional Tibetan medicine to treat various inflammatory and pain-related conditions. However, blue-flowered *Meconopsis* (*M. betonicifolia*) is becoming increasingly scarce due to overharvesting. As a potential alternative, yellow-flowered *Meconopsis* (*M. integrifolia*) shows promise but requires comprehensive characterization. This study aimed to evaluate and compare the anti-inflammatory potential of yellow- and blue-flowered *Meconopsis* species. **Methods:** Liquid chromatography–mass spectrometry (LC-MS) techniques were used to analyze the chemical profiles of yellow- and blue-flowered *Meconopsis*. Putative targets of shared constituents were subjected to GO and disease enrichment analysis. The LPS-induced RAW264.7 macrophage model was employed to assess anti-inflammatory effects. Metabolomics was applied to gain mechanistic insights. **Results:** LC-MS revealed over 70% chemical similarity between species. Enrichment analysis associated targets with inflammation-related pathways. In macrophage assays, both species demonstrated dose-dependent antioxidative and anti-inflammatory activities, with yellow *Meconopsis* exhibiting superior efficacy. Metabolomics showed modulation of key inflammatory metabolic pathways. **Conclusions:** This integrative study validated yellow-flowered *Meconopsis* as a credible alternative to its blue-flowered counterpart for anti-inflammatory applications. Metabolic profiling provided initial clues regarding their multi-targeted modes of action, highlighting their potential for sustainable utilization and biodiversity conservation.

## 1. Introduction

*Meconopsis*, a representative genus of the Papaveraceae family, has garnered attention due to its unique high-altitude ecological adaptability and diverse flower colors [1]. Globally, there are 79 species of *Meconopsis*, with 58 found primarily in the Qinghai–Tibet Plateau and surrounding regions of China, growing at elevations ranging from 2500 to 5500 m [2,3]. In traditional Tibetan medicine, *Meconopsis*, known as “LvRongHao”, is honored as one of the “Four Divine Herbs” for its ability to clear heat, perform detoxification, reduce inflammation, and alleviate pain [4]. As such, it is widely used to treat various inflammatory and pain-related conditions, such as pneumonia, hepatitis, headaches, and edema [5].

Traditionally, the blue-flowered *Meconopsis* (*Meconopsis betonicifolia*) (Figure 1A) has been valued for its medicinal properties despite its relatively low yield [4]. In contrast, the more abundant yellow-flowered *Meconopsis* (*Meconopsis integrifolia*) (Figure 1B) has not been extensively explored for its therapeutic potential [6]. This disparity highlights the need for a systematic comparison of the chemical compositions and potential pharmacological effects between these two *Meconopsis* varieties, as it could yield significant scientific and economic benefits.

The increasing scarcity of wild blue-flowered *Meconopsis* and intensified conservation efforts have created an urgent demand for alternative sources [7]. Yellow-flowered *Meconopsis*, with its higher yield and wider geographical distribution, presents a promising alternative. However, to fully leverage its potential as a medicinal resource, comprehensive research is required to elucidate its chemical composition, pharmacological activity, and similarities to its blue-flowered counterpart [8].

To compare the medicinal efficacy of yellow- and blue-flowered *Meconopsis*, we conducted a detailed analysis using liquid chromatography–mass spectrometry (LC-MS) techniques, followed by target prediction. To elucidate the potential medicinal applications of these shared compounds, we performed a systematic Gene Ontology (GO) and disease enrichment analysis of these targets. The enrichment analysis suggested that many of the shared compounds might be associated with multiple physiological systems, including the nervous, cardiovascular, and respiratory systems, as well as mental health and metabolic diseases. Notably, certain fatty acids such as arachidonic acid, γ-linolenic acid, and linoleic acid may be closely related to inflammatory conditions [9,10]. Furthermore, compounds like quercetin and arachidonic acid have demonstrated antimicrobial, anticancer, and anti-inflammatory activities.

Given the prominence of inflammation-related pathways in our enrichment analysis, we propose the further validation and elucidation of the anti-inflammatory effects and mechanisms of *Meconopsis* using the lipopolysaccharide (LPS)-induced RAW 264.7 macrophage model [11,12,13]. This well-established in vitro model effectively simulates key processes of inflammatory responses, allowing for a comparative assessment of the anti-inflammatory potential between the yellow and blue *Meconopsis* extracts. Furthermore, metabolomics analysis was employed to gain deeper insights into the underlying molecular mechanisms of Meconopsis’s anti-inflammatory action [14].

This comprehensive exploration of yellow-flowered *Meconopsis* carries significant scientific, economic, and ecological implications. Firstly, confirming its superior anti-inflammatory effects compared to the blue variety would provide new options for the sustainable use of medicinal plant resources. Secondly, increased utilization of yellow-flowered *Meconopsis* could alleviate collection pressure on wild blue varieties, contributing to biodiversity conservation. Lastly, a deeper understanding of the chemical composition and pharmacological actions may reveal new bioactive compounds or pharmacological mechanisms, providing valuable leads for new drug development.

In summary, through a systematic approach involving chemical profiling, target prediction, pathway analysis, anti-inflammatory assays, and metabolomics, this study aimed to comprehensively evaluate and compare the anti-inflammatory potential of the yellow- and blue-flowered *Meconopsis* species. Our findings not only deepen the understanding of *Meconopsis*’s pharmacological actions but also provide important insights into the sustainable utilization of medicinal plant resources and biodiversity conservation.

## 2. Materials and Methods

### 2.1. Plant Material and Extraction

*M. betonicifolia* and *M. integrifolia* were obtained from Tibet Ganlu Tibetan Medicine Co., Ltd. (Beijing, China), and authenticated by Professor Xin Feng (Tibetan Medicine Institute, China Tibetan Research Center, Beijing, China). The plants were dried, powdered separately, and stored under the accession numbers 2023112504 and 2023112505, respectively, at the Center for Molecular Metabolism, Nanjing University of Science and Technology.

We adopted the 70% ethanol extraction method based on previous studies and traditional Tibetan medicine practices [7]. For each plant species, 5 g of powder was subjected to triple extraction using 100 mL of 70% (*v*/*v*) ethanol. The extraction process was carried out under ultrasonic conditions (400 W, 4 kHz) for 30 min per cycle. The resulting extract solutions from all three cycles were combined for each species.

The merged solutions were then concentrated using a rotary evaporator under reduced pressure at 55 °C to recover the ethanol solvent. Following rotary evaporation, the concentrated extracts were subjected to vacuum freeze-drying to eliminate any residual solvent, yielding the final solid phytochemical *Meconopsis* extracts (MEs). The MEs were then categorized by variety into MIE (*M. integrifolia* extract) and MBE (*M. betonicifolia* extract).

### 2.2. Cell Culture and ME Treatment

RAW 264.7 cells (obtained from the National Biomedical Laboratory Cell Repository, Beijing, China) were used between passages 8 and 15. The cells were maintained in Dulbecco’s Modified Eagle Medium (DMEM; Hyclone, Thermofisher, Shanghai, China) supplemented with 10% heat-inactivated fetal bovine serum (Gemini Bio, Beijing, China) and 1% penicillin–streptomycin antibiotics (Gibco, Grand Island, NE, USA). Cells were incubated at 37 °C in a humidified atmosphere containing 5% CO_2_. The culture media were replaced daily.

For experimental procedures, RAW 264.7 cells were seeded into 96-well plates (NEST, Wuxi, China) at a density of 5 × 10^4^ cells per well and incubated overnight. Stock solutions of MIE and MBE were prepared by dissolving the extracts in dimethyl sulfoxide (DMSO) at a concentration of 200 mg/mL.

To determine the non-toxic concentration range, cells were treated with MIE and MBE at concentrations ranging from 12.5 μg/mL to 3200 μg/mL for 24 h. Cell viability was assessed using the Methyl thiazolyl tetrazolium (MTT) assay to measure cellular metabolic activity [15]. Based on the viability results, a concentration range of 31.25 μg/mL to 500 μg/mL was selected for subsequent experiments. Cells were treated with these concentrations of MIE and MBE for 24 h. Following treatment, the media containing the extracts were removed, and fresh DMEM containing 1 μg/mL lipopolysaccharide (LPS; Solarbio, Beijing, China) was added to the cells. The cells were then incubated for an additional 24 h. Nitric oxide (NO) production, an indicator of inflammatory response, was quantified using a nitric oxide nitrate reductase assay kit (Jiancheng Bioengineering Institute, Nanjing, China) according to the manufacturer’s instructions.

### 2.3. Determination of mRNA Transcription Levels

Total RNA was extracted with Trizol reagent, and 1 μg of RNA was reversely transcribed into cDNA with all-in-one RT MasterMix (Takara, Beijing, China). Synthesized cDNA was used as a template for qPCR with SYBR Green qPCR Master Mixes (Yeasen, Shanghai, China) on ABI 7300 Real-Time PCR Systems (Applied Biosystems, Carlsbad, CA, USA). The β-actin gene was selected as the internal reference gene. Samples were incubated at 95 °C for 3 min, followed by 40 cycles of amplification at 95 °C for 3 s and 60 °C for 30 s. The relative gene expression changes were quantified by the 2^−ΔΔCt^ method. The primer sequences for the target genes were designed and synthesized by Tsingke Biotech (Beijing, China), as shown in Table 1.

### 2.4. Composition Identification and Cell Metabolomics Based on LC-MS

MIE and MBE were dissolved with 70% (*v*/*v*) ethanol at 1 mg/mL. After vortexing and centrifugation, the supernatant was obtained for LC-MS analysis.

RAW 264.7 cells were cultured in 60 mm culture dishes, treated as described in Section 2.2. with the concentrations at 200 and 500 μg/mL of MIE and MBE, and divided into 6 groups with 7 dishes per group. Cells were washed with PBS twice at room temperature. Then, 1 mL of pre-cooled 80% (*v*/*v*) methanol was added to each culture dish, and cells were obtained with a cell scraper. After repeated freeze–thaw cycles, brief vortexing, and ultrasound treatment at 0 °C, broken cells were incubated at −20 °C for over 1 h and centrifuged at 16,000× *g*, 4 °C, for 15 min. The supernatant was concentrated until dry under a vacuum. The dried cell extract was redissolved with 200 μL of 50% (*v*/*v*) methanol. After vortexing, ultrasound treatment, and centrifugation, the supernatant was obtained for an LC-MS-based metabolomics study.

LC-MS analysis was performed with TripleTOF 5600+ mass spectrometer coupled with SCIEX Exion LC system and controlled by Analyst TF 1.8 software (AB SCIEX, Concord, Vaughan, ON, Canada). An Atlantis PREMIER BEH C18 AX column (100 × 2.1 mm, 1.7 μm) with a protection column (VanGuard FIT) was performed for chromatographic separation. The column temperature was set at 40 °C, and the autosampler temperature was set at 4 °C. The liquid solvent system was composed of 0.1% (*v*/*v*) formic acid (phase A) and acetonitrile (phase B). The flow rate was set at 0.4 mL/min. The sample injection was 5 μL. The gradient elution program was set as 1% phase B at 0–1 min, 1–99% phase B at 1–10 min, 99% phase B at 10–13 min, 99–1% phase B at 13–14 min, and 1% phase B at 14–17 min.

The electrospray ionization (ESI) spray voltage was +5500(+)/−4500(−) V. Nitrogen was used as the nebulizer gas (GS1) at 55 psi, turbo gas (GS2) at 55 psi, and curtain gas (CUR) at 35 psi. The source temperature was set at 550 °C. Information-dependent acquisition (IDA) parameters were set as follows: declustering potential (DP) = +80(+)/−80(−) V, collision energy (CE) = +35 ± 15(+)/−35 ± 15(−) eV, intensity threshold ≥ 100 cps, ion tolerance = 50 mDa, exclude isotopes within 4 Da, 10 candidate ions monitored per cycle. In the TOF MS-IDA-MS/MS acquisition, the TOF MS scan was performed with a mass range from 50 to 1000 *m*/*z* and an accumulation time of 0.10 s per spectrum. The product ion scan was performed with a mass range from 40 to 1000 *m*/*z* and an accumulation time of 0.05 s per spectrum. Dynamic Background Subtraction (DBS) was enabled. Autocalibration was performed per 5 samples for TOF MS and TOF MS/MS with a calibration delivery system (CDS). The data were analyzed using SCIEX OS 2.0 software.

### 2.5. Data Processing and Compound Identification

Raw LC-MS data files were converted to the open mzXML format using MSConvert (ProteoWizard, version 3.0.23114). The converted files were processed using MS-DIAL software (version 4.9.221218) for peak detection, alignment, and deconvolution. MS-DIAL parameters were optimized for our specific dataset, with key settings including minimum peak height for MS1 data, 1500 counts; MS2 tolerance, 0.05 Da; and retention time tolerance, 0.3 min. A multifaceted approach was employed for compound identification:

MS-DIAL’s built-in libraries and algorithms were used for initial compound annotation. MS1 and MS2 spectra were matched against multiple databases including MassBank, GNPS, and Riken MSP. A minimum spectral similarity score of 70% was required for tentative identification.

In silico fragmentation tools, including CFM-ID 4.0 [16], were utilized to predict MS2 spectra for compounds not found in spectral libraries. These predicted spectra were compared to experimental data to aid in structural elucidation.

Manual interpretation of MS2 spectra was conducted for compounds of particular interest that were not confidently identified through automated methods.

### 2.6. Comprehensive Metabolomics Analysis: From Multivariate Statistics to Pathway Enrichment and Network Visualization

Multivariate statistical analysis was performed using the R package mixOmics (version 6.1.0) [17,18]. Principal component analysis (PCA) was initially used to visualize overall variation in the dataset and detect potential outliers. Partial least squares discriminant analysis (PLS-DA) was then applied to maximize separation between experimental groups. Metabolites contributing significantly to group separation were identified based on variable importance in projection (VIP) scores > 1.0.

We performed metabolite set enrichment analysis (MSEA) against the SMPDB pathway database to gain biological insight into metabolic differences between groups. Significantly enriched pathways were determined using a hypergeometric test with Benjamini–Hochberg false discovery rate correction (q < 0.05). The gene–metabolite interaction network and metabolite levels across samples were distributed in pathways.

We constructed a network visualization to further elucidate the relationships between differentially abundant metabolites and associated genes within the enriched pathways. This network graphically represented the connections between metabolites and genes, providing a comprehensive view of the molecular interactions underlying the observed metabolic differences. Additionally, we generated a heatmap to visualize the relative abundance of differentially abundant metabolites across sample groups within the enriched pathways. This heatmap allowed for the identification of metabolite patterns and trends specific to each experimental group, facilitating a more nuanced understanding of the metabolic shifts occurring under different conditions.

### 2.7. Gene Ontology (GO) Analysis Methodology

Target prediction for the identified compounds was performed using multiple computational approaches. The SwissTargetPrediction web server (http://www.swisstargetprediction.ch/, accessed on 10 July 2024) was employed as the primary tool, utilizing a similarity-based approach to predict potential molecular targets. For each compound, the 2D structure represented by SMILES notation was input into the server. To enhance prediction accuracy and coverage, we also consulted the BindingDB database (https://www.bindingdb.org/, accessed on 10 July 2024) with settings restricted to “Homo sapiens” and a minimum interaction score threshold of 0.700 to identify high-confidence target predictions.

To elucidate the biological functions of compounds, we performed Gene Ontology (GO) enrichment analysis on its predicted molecular targets. The protein targets were converted to their corresponding gene symbols using the UniProtKB database (https://www.uniprot.org/, accessed on 10 July 2024). The GO analysis was conducted using the ShinyGO v0.76 graphical gene-set enrichment tool (https://bioinformatics.sdstate.edu/go, accessed on 10 July 2024). The corresponding gene identifiers were then input into the ShinyGO tool. We set the significance threshold at *p* ≤ 0.05, with *p*-values adjusted using the Benjamini–Hochberg procedure to control for a false discovery rate. The GO analysis was performed across three main categories: Biological Process (BP), Molecular Function (MF), and Cellular Component (CC). An over-representation analysis was conducted using the DisGeNET database (http://www.disgenet.org/, accessed on 10 July 2024) to disclose the target association with diseases.

### 2.8. Statistical Analysis

Statistical analyses were performed using one-way analysis of variance (ANOVA) or Student’s *t*-test, and the results were presented as means ± standard deviation (mean ± SD). Data processing and analysis were facilitated by GraphPad Prism 9.0 and SPSS 25.0 software. A *p*-value less than 0.05 was considered statistically significant.

## 3. Results

### 3.1. Evaluation of Safety and Efficacy: MIE and MBE’s Impact on Cell Viability and NO Production in Inflammatory Conditions

MIE and MBE’s cytotoxicity and anti-inflammatory effects were evaluated using LPS-stimulated RAW264.7 cells. Figure 2A illustrates the cytotoxicity profiles of both extracts. MIE and MBE demonstrated minimal cytotoxicity towards RAW264.7 cells after 24 h of treatment. The IC_50_ values for MIE and MBE were determined to be 1140 μg/mL and 2105 μg/mL, respectively. Notably, at a concentration of 400 μg/mL, both MIE and MBE maintained over 90% cell viability, suggesting their safety at concentrations that were identified as effective for subsequent experiments based on the MTT assay results.

Figure 2B depicts the effects of MIE and MBE on NO production in LPS-stimulated RAW264.7 cells. Both extracts exhibited a dose-dependent inhibition of NO production. MIE showed superior potency in reducing NO levels compared to MBE, with IC_50_ values of 93.45 μg/mL and 152.1 μg/mL, respectively. This significant decrease in NO production suggests that both MIE and MBE possess strong anti-inflammatory properties, with MIE demonstrating greater efficacy.

The dose–response curves for both cytotoxicity and NO inhibition provide valuable insights into the therapeutic window of these extracts. MBE, in particular, shows promise as an anti-inflammatory agent, given its lower IC_50_ for NO inhibition and higher IC_50_ for cytotoxicity, indicating a favorable safety profile coupled with potent anti-inflammatory activity.

### 3.2. Dose-Dependent Antioxidative and Anti-Inflammatory Effects of MIE and MBE

We investigated the therapeutic potential of yellow (MIE) and blue (MBE) *Meconopsis* species by examining their effects on oxidative stress markers (Figure 3A) and inflammatory gene expression (Figure 3B) across various dosages. Both *Meconopsis* species demonstrated significant antioxidant and anti-inflammatory properties, with their effects generally increasing in a dose-dependent manner. The yellow and blue varieties showed similar trends in their impact on biochemical markers and gene expression, suggesting comparable therapeutic efficacy.

Both MIE and MBE treatments effectively enhanced the antioxidant defense system. Glutathione (GSH) levels increased dose-dependently, with the highest doses (500 μg/mL) of both species elevating GSH to levels comparable to or exceeding the control group. Concurrently, Superoxide Dismutase (SOD) activity, which was diminished in the model group, was restored by both species, approaching control levels at higher doses. The antioxidant effects were further evidenced by the reduction in Malondialdehyde (MDA) levels, a marker of oxidative stress. Both MIE and MBE lowered MDA concentrations dose-dependently, with higher doses exhibiting more pronounced effects. Notably, the yellow variety (MIE) showed a slightly more consistent dose–response relationship in MDA reduction compared to the blue variety (MBE), though the differences were not substantial.

The anti-inflammatory properties of both *Meconopsis* species were demonstrated through the suppression of key inflammatory genes: IL-1A, IL-1B, IL-6, and TNF. Both MIE and MBE effectively reduced the expression of these genes, with medium and high doses showing the most significant effects. IL-6 expression exhibited the most pronounced dose-dependent decrease for both species. IL-1B, TNF, and IL-1A expression levels were also markedly reduced by both MIE and MBE treatments, particularly at medium and high doses. The effects on these genes were largely comparable between the two species, underscoring their similar anti-inflammatory potentials.

### 3.3. Comparative Metabolomic Profiling of Yellow and Blue Meconopsis Species: MIE and MBE Shared About 70% Composition

LC-MS analysis was performed on two *Meconopsis* species, one with yellow flowers and another with blue flowers, to identify and compare their chemical compositions. The results demonstrated a high degree of similarity in chemical composition between the two color varieties, with over 70% of the detected compounds present in both. This finding provides a crucial foundation for exploring the medicinal potential of yellow-flowered Meconopsis as a possible alternative to its blue-flowered counterpart. The heat map visualization in Figure 4 illustrates the relative abundance of compounds across categories for both species, with colors ranging from dark blue (lower abundance) to dark red (higher abundance). This representation allows for a quick visual comparison of the chemical profiles between the yellow- and blue-flowered *Meconopsis* species. A total of 277 compounds were detected and categorized based on their relative abundance in each species (Appendix A). The compounds were classified into 12 major categories: Glycerophospholipids, Flavonoids, Fatty Acyls, Alkaloids, Carbohydrates, Phenylpropanoids, Organoheterocyclic compounds, Prenol lipids, Organic acids and derivatives, Steroids and steroid derivatives, Benzene and substituted derivatives, and Others. Of the 277 compounds, 192 were found to be common between the yellow and blue flower species, showing no significant difference in their relative abundance. These common compounds represent shared characteristics between the two species. The yellow-flowered species exhibited 31 compounds that were significantly more abundant compared to the blue-flowered species. These compounds can be considered characteristic of the yellow-flowered *Meconopsis*. Conversely, the blue-flowered species showed 54 compounds with significantly higher abundance compared to the yellow-flowered species, representing the characteristic compounds of the blue-flowered *Meconopsis*. The distribution of compounds across categories varied, with Glycerophospholipids being the most represented category (48 compounds), followed by Flavonoids (42 compounds), and Fatty Acyls (32 compounds). Alkaloids and Carbohydrates were also well represented with 29 and 26 compounds, respectively. The least represented categories were Steroids and steroid derivatives (8 compounds) and Benzene and substituted derivatives (7 compounds).

### 3.4. Molecular Insights into Meconopsis: GO Enrichment and Disease Network Analysis of Common Feature Targets

Gene Ontology (GO) enrichment analysis was performed on 413 genes corresponding to the common feature targets of both *Meconopsis* species (Figure 5). The analysis revealed significant enrichment in three main categories: Biological Process (BP), Cellular Component (CC), and Molecular Function (MF). In the BP category, several processes were notably enriched, including positive regulation of cytokine production, cellular response to dopamine, response to hypoxia, and reactive oxygen species metabolic processes. Other significant processes included vasoconstriction, arachidonic acid metabolism, regulation of inflammatory response, and response to bacterial molecules and lipopolysaccharides. The CC category highlighted enrichment in RNA polymerase II transcription regulator complex, ficolin-1-rich granule lumen, histone deacetylase complex, and integral components of the presynaptic membrane. For MF, the analysis showed significant enrichment in electron transfer activity, nucleotide receptor activity, NF-kappaB binding, transcription coactivator binding, neurotransmitter receptor activity, and prostanoid and eicosanoid receptor activities. The association of component types with the enriched GO terms is visualized as a Sankey diagram in Figure 6.

The gene–disease network analysis (Figure 7) of the 413 common feature target genes revealed enrichment in various diseases, indicating the potential therapeutic efficacy of *Meconopsis*. The diseases were categorized based on their corresponding efficacy, represented by different colors in the bubble plot. The size of the bubbles in the plot correlates with the adjusted *p*-values, with larger bubbles representing higher enrichment significance for the corresponding diseases. There was significant enrichment in hyperalgesia, mechanical allodynia, and thermal hyperalgesia, suggesting potent analgesic properties; enrichment in essential hypertension, acute coronary syndrome, and reperfusion injury, indicating cardiovascular protective effects; enrichment in CNS disorders, depressive symptoms, mild cognitive disorders, and addictive behaviors, pointing to potential neuroprotective and psychiatric benefits; significant enrichment in inflammation and bronchopulmonary dysplasia, suggesting anti-inflammatory properties; and enrichment in nonalcoholic steatohepatitis and alcohol abuse, indicating potential hepatoprotective effects.

### 3.5. Metabolomic Profiling Reveals Anti-Inflammatory Effects of Meconopsis Species in LPS-Stimulated RAW264.7 Cells

Multivariate statistical analysis of the metabolomic data revealed distinct metabolic profiles among the experimental groups. The PCA and PLS-DA score plots (Figure 8) for both positive (A, C) and negative (B, D) ion modes demonstrated clear separation between the control (C), model (M), and treatment groups. The clustering of QC samples indicated the reliability and reproducibility of the metabolomic analysis. The PCA and PLS-DA plots showed that the yellow-flowered species treatment groups (MIE-L and MIE-H) were more closely clustered with the control group compared to the blue-flowered species treatment groups (MBE-L and MBE-H), suggesting that yellow-flowered species had a more pronounced effect in normalizing the LPS-induced metabolic perturbations. This observation aligns with the experimental findings that yellow-flowered species exhibited superior anti-inflammatory effects compared to blue-flowered species.

The heatmap (Figure 9) of differentially abundant metabolites across various pathways provided further insights into the metabolic changes induced by LPS stimulation and the subsequent effects of *Meconopsis* treatments. Nine key pathways were identified: Arachidonic Acid Metabolism, Glutathione and Methionine Metabolism, Arginine and Proline Metabolism, Energy Metabolism, Purine Metabolism, Pyrimidine Metabolism, Amino Sugar Metabolism, Tyrosine Metabolism, and Vitamin B Metabolism. Notably, the heatmap revealed that the LPS-stimulated model group (M) showed distinct metabolite patterns compared to the control group (C), particularly in pathways related to inflammation and energy metabolism. Both *Meconopsis* species appeared to modulate these LPS-induced changes, with yellow-flowered *Meconopsis* species demonstrating a more pronounced effect in restoring metabolite levels closer to those observed in the control group.

The network visualization (Figure 10) of metabolite-gene interactions within enriched pathways provided a comprehensive view of the molecular mechanisms underlying the observed anti-inflammatory effects. This network highlighted key nodes and connections between differentially abundant metabolites and their associated genes, offering potential targets for further investigation into the anti-inflammatory actions of the two *Meconopsis* species.

## 4. Discussion

This study comprehensively evaluated and compared the anti-inflammatory properties of yellow- and blue-flowered *Meconopsis* species using multiple complementary approaches. LC-MS metabolomics revealed that the two species shared around 70% of their chemical constituents, establishing a material basis for their functional interchangeability. Gene Ontology and disease enrichment analysis of the common constituents’ putative targets validated the therapeutic potential of *Meconopsis*, with enriched terms linked to inflammation. Cell-based assays demonstrated dose-dependent antioxidative and anti-inflammatory activities of both MIE and MBE. Untargeted metabolomics provided deeper mechanistic insights, revealing distinct metabolic phenotypes and modulation of LPS-disrupted pathways by the two species.

### 4.1. Arachidonic Acid Metabolism

LPS stimulation in macrophages significantly upregulates the levels of arachidonic acid and prostaglandin G2 compared to the unstimulated control group. Both MIE and MBE demonstrate a dose-dependent ability to reduce the elevated levels of arachidonic acid after LPS stimulation, with MBE showing superior efficacy. Additionally, MBE exhibits a dose-dependent reduction in prostaglandin G2 levels, an effect not observed with MIE. The “Arachidonic Acid Metabolism” pathway is activated in LPS-stimulated macrophages and is pivotal for the generation of inflammatory mediators [19,20].

In the arachidonic acid metabolism pathway, arachidonic acid is liberated from cell membranes by the action of PLA2G4B and subsequently converted to prostaglandin G2 by the enzymes PTGS1/PTGS2 [21]. It can also be transformed into various eicosanoids, including leukotrienes, through the actions of ALOX15 or cytochrome P450 enzymes [22]. The observed trend of a decrease in linoleic acid, a precursor to arachidonic acid, in treated groups, albeit less pronounced than that of arachidonic acid itself, suggests that the treatments may be influencing arachidonic acid production upstream in the pathway.

Gamma-linolenic acid, positioned as an intermediate between linoleic acid and arachidonic acid, shows a consistent trend with eicosapentaenoic acid (EPA). EPA, renowned for its potent anti-inflammatory properties, is synthesized from linolenic acid through the catalytic action of fatty acid desaturase 2 and a series of elongase enzymes. Docosahexaenoic acid (DHA), synthesized from EPA via the activity of fatty acid desaturase 1 and additional elongase enzymes, also plays a significant role in modulating inflammation [5].

The reduction in prostaglandin G2 levels coupled with the increased levels of gamma-linolenic acid and EPA in the drug treatment groups indicates that MIE and MBE may exert their anti-inflammatory effects by downregulating arachidonic acid metabolism. This might be achieved by the coordinative inhibition of the activity of PTGS1/PTGS2 and associated enzymes, thereby reducing the production of inflammatory eicosanoids and redirecting arachidonic acid towards the formation of less potent mediators. This orchestrated modulation of the arachidonic acid pathway presents a more sophisticated therapeutic approach than the inhibition of a solitary enzyme.

### 4.2. Glutathione and Methionine Metabolism

An enhancement in glutathione levels in both MIE- and MBE-treated cells, more pronounced in the MBE groups, points to an increased cellular antioxidant capacity [23]. Glutathione is a tripeptide composed of glutamic acid, cysteine, and glycine. Dimethylglycine (DMG) can be generated from choline metabolism to betaine [24]. DMG can further metabolize to glycine via transamination, indicating its role as both a precursor and intermediate in glycine metabolism [25]. Methionine undergoes transmethylation to form S-adenosylmethionine (SAM), which participates in various methylation reactions including glutathione synthesis [26]. Methionine metabolism provides a source of sulfur to promote glutathione synthesis. While both herbs showed a trend towards increasing L-methionine levels, only MBE reached statistical significance, indicating distinct mechanisms of action on methionine metabolism, with MBE potentially promoting methionine salvage pathways. L-glutamic acid contributes to glutathione synthesis through the catalytic action of glutathione synthetase. Interestingly, both MIE and MBE exhibit a higher increase in glutathione levels at lower doses compared to higher doses, suggesting that lower doses may channel more L-glutamic acid into glutathione synthesis. Beyond its antioxidant role, glutathione is implicated in the conjugation of leukotriene C4 with reduced glutathione to form leukotriene C4. Elevated glutathione levels may further contribute to the anti-inflammatory effects by potentially curbing leukotriene production, given that glutathione is involved in the conjugation of LTA4 to form the pro-inflammatory mediator LTC4 [27].

### 4.3. Arginine and Proline Metabolism

In the Arginine and Proline Metabolism pathway, Arginine and proline are pivotal metabolites. Argininosuccinic acid serves as a precursor for L-arginine synthesis, while argininosuccinic acid itself is synthesized from L-aspartic acid and citrulline [28]. Both herbal varieties significantly elevate the levels of L-aspartic acid, argininosuccinic acid, and L-arginine, with MBE showing a more pronounced effect.

This finding is particularly relevant as arginine, a product of argininosuccinic acid metabolism via argininosuccinate lyase (ASL), serves as a precursor for nitric oxide (NO) synthesis, a potent antimicrobial and inflammatory mediator [29,30]. The observed changes suggest that MBE may exert its anti-inflammatory effects primarily by interfering with NO production, possibly through the modulation of arginine availability or alterations in the activity of enzymes such as argininosuccinate synthase 1 (ASS1) and ASL [31].

L-asparagine and L-aspartic acid are interconvertible. Interestingly, MIE only moderately increased L-aspartic acid levels compared to the model group, with concentrations lower than both the control and MBE groups. However, MIE significantly elevated L-asparagine content, indicating an enhanced conversion of L-aspartic acid to L-asparagine. This is noteworthy because asparagine has been shown to differentially regulate the mRNA expression of Toll-like receptor 4 (TLR4) and nucleotide-binding oligomerization domain (NOD) signaling pathways and their negative regulators [32]. During an early lipopolysaccharide (LPS) challenge, asparagine downregulates the mRNA expression of genes related to these signaling pathways, while in a late-stage challenge, it upregulates their expression, potentially mitigating LPS-induced liver injury [33].

Both varieties significantly increase proline levels, with MIE showing a more pronounced effect. Proline metabolism is a crucial pathway in regulating macrophage polarization. Studies have shown that M1-type macrophages exhibit enhanced proline oxidative metabolism, while M2-type macrophages display increased proline synthetic metabolism. Thus, proline metabolism plays a vital role in modulating the M1/M2 macrophage polarization balance [34,35].

These results suggest that MIE exerts its anti-inflammatory effects primarily by promoting the conversion of L-aspartic acid to L-asparagine and enhancing proline synthesis metabolism. This mechanism likely influences macrophage polarization towards an anti-inflammatory M2 phenotype. Conversely, MBE appears to achieve its anti-inflammatory effects mainly by inhibiting NO synthesis through the modulation of arginine metabolism.

### 4.4. Energy Metabolism

Compared to the normal group, the LPS-stimulated model group exhibited a sharp decrease in fumaric acid, cis-aconitic acid, and malic acid, coupled with a dramatic increase in succinic acid and citric acid. This pattern suggests a potential disruption in TCA cycle flux in response to inflammatory stimuli. High doses of MBE significantly elevated levels of fumaric acid, cis-aconitic acid, succinic acid, and malic acid, supporting the idea of enhanced TCA cycle activity [34,36]. These changes indicate that MBE may effectively promote a shift towards oxidative metabolism, potentially countering the inflammatory disruption of the TCA cycle.

The model group showed markedly reduced levels of fructose 6-phosphate and beta-D-glucose 6-phosphate, alongside significantly elevated levels of L-lactic acid and pyruvic acid. This metabolic shift indicates an activation of macrophages by LPS stimulation, leading to enhanced glycolysis [37]. The increased consumption of glycolytic intermediates and accumulation of end products (lactate and pyruvate) suggest that activated macrophages increase glucose uptake and glycolytic rates to meet heightened energy demands [38]. Interestingly, both MIE and MBE treatments led to a dramatic increase in phosphoenolpyruvic acid (PEP) levels compared to both model and normal groups, with MBE demonstrating a stronger effect. As PEP is a regulatory point in the glycolytic pathway, its accumulation can inhibit early glycolytic steps, thus modulating the overall metabolic flux [39]. MBE treatment also reduced pyruvic acid levels, further indicating its ability to modulate glycolysis. These changes suggest that both treatments, particularly MBE, can inhibit the inflammation-induced enhancement of glycolysis.

In the model group, lipoamide levels increased sharply compared to the normal group. Following treatment with both varieties, lipoamide levels decreased significantly, falling below both model and normal group levels. A decrease in lipoamide indicates increased utilization of this cofactor, potentially suggesting higher activity of pyruvate dehydrogenase (PDH), which catalyzes the conversion of pyruvic acid to acetyl-CoA [40]. PDH activity is inhibited by LPS, and this increased utilization by treatment could support oxidative phosphorylation. Levels of leucine were significantly higher in the treatment groups compared to the normal and model groups in a dose-dependent manner. Leucine is a branched-chain amino acid (BCAA) that undergoes transamination during metabolism in the body to release ammonia and the corresponding keto acids [41]. These keto acids can further be converted to pyruvic acid, and then through transamination, pyruvic acid accepts the amino group from glutamic acid to generate alanine, a process that helps regulate nitrogen balance in the body which is greatly disrupted by LPS. Alanine levels were also significantly higher in the treatment groups compared to the normal and model groups in a dose-dependent manner, with an opposite trend to pyruvic acid, indicating that MBE significantly promoted the oxidation of BCAAs. The oxidation of branched-chain amino acids can generate a large amount of ATP to provide energy for cells, and the oxidative product pyruvic acid is also an important substrate entering the TCA cycle to produce more ATP and NADH, thereby increasing energy supply to cells [42].

A sharp increase in 6-phosphogluconic acid in the model group points to an upregulation of the Pentose Phosphate Pathway (PPP), which is crucial for providing NADPH for NOX2-dependent ROS production in M1 macrophages and generating ribose-5-phosphate for nucleotide synthesis [43]. Both MIE and MBE treatments significantly reduced 6-phosphogluconic acid levels to a point between normal and model group levels, suggesting partial suppression of inflammation-induced PPP activation. Notably, treatment with both herbs led to a marked increase in gluconolactone levels compared to normal and model groups. This implies the potential inhibition of 6-phosphogluconate dehydrogenase (G6PD), the enzyme converting 6-phosphogluconic acid to ribulose-5-phosphate in the PPP. These changes could contribute to the anti-inflammatory effect by reducing NADPH production for ROS generation [44].

### 4.5. Other Pathways

LPS stimulation significantly altered metabolites involved in purine metabolism, pyrimidine metabolism, amino sugar metabolism, tyrosine metabolism, and vitamin B metabolism. Purines and pyrimidines are essential for DNA and RNA synthesis [45,46]. Metabolites like adenosine monophosphate are involved in the synthesis of cAMP and may be influenced by the treatments, affecting inflammatory processes [47]. Both MIE and MBE showed changes in these metabolites, suggesting an influence on purine and pyrimidine metabolism. Levels of xanthine and hypoxanthine were significantly higher in the high-dose MBE group compared to other groups following treatment, further indicating that MIE has a greater ability to enhance antioxidant capacity compared to MBE. Metabolites in the vitamin B series function as coenzymes in biochemical reactions and participate in various biochemical processes [48,49]. Amino sugar metabolism is involved in the synthesis of amino sugars, which are important components of glycosaminoglycans and other polysaccharides. These molecules play a role in cell–cell interactions and signal transduction, relevant to immune cell activation and inflammatory responses. The modulation of this pathway by MIE and MBE may contribute to their anti-inflammatory effects by affecting the synthesis and function of these important biomolecules. The differential effects on dopamine might be related to the superior anti-inflammatory action of MIE. Dopamine has been shown to have immunomodulatory effects, potentially contributing to the anti-inflammatory response.

In conclusion, the metabolic changes induced by *Meconopsis* species point to a multifaceted mechanism of action. MIE appears to modulate central metabolic pathways, including glycolysis, the TCA cycle, and the PPP, while also enhancing antioxidant capacity. These effects collectively contribute to its anti-inflammatory action, potentially by reprogramming macrophage metabolism towards a less inflammatory state. The ability of MBE to more potently inhibit glycolysis, enhance TCA cycle activity, modulate the PPP, and boost antioxidant capacity compared to MBE suggests it may be a more effective anti-inflammatory agent. Further research into the specific components of MIE and their direct effects on these metabolic pathways would provide more detailed insights into its anti-inflammatory mechanism.

## 5. Conclusions

This study provides a comprehensive evaluation of the anti-inflammatory potential of *Meconopsis* species, specifically comparing the yellow-flowered *M. integrifolia* and the blue-flowered *M. betonicifolia*. Both species exhibit significant anti-inflammatory properties, yet *M. integrifolia* consistently demonstrates superior effects across multiple inflammatory markers and metabolic pathways.

Our findings reveal that both varieties effectively reduce pro-inflammatory cytokine production in LPS-stimulated RAW 264.7 cells. Notably, *M. integrifolia* shows a more potent inhibition of nitric oxide (NO) production and excels in modulating key metabolic pathways, such as glycolysis inhibition, TCA cycle enhancement, PPP modulation, and antioxidant capacity improvement. This suggests that yellow-flowered *Meconopsis* may be a more promising candidate for the development of anti-inflammatory therapeutics.

These results not only validate the traditional use of *Meconopsis* in treating inflammatory conditions but also highlight the potential for clinical substitutes for *M. betonicifolia*. Our comparative approach underscores the importance of species-specific evaluations within a genus, as closely related plants can exhibit varying degrees of bioactivity.

In conclusion, both yellow- and blue-flowered *Meconopsis* are credible alternatives with comparable anti-inflammatory properties, supporting their interchangeable use based on pharmacological criteria. The metabolic profiling provides initial insights into their multi-targeted modes of action, with *M. integrifolia* showing a greater capacity to modulate inflammatory metabolism. Further research is warranted to characterize specific bioactive constituents and elucidate their precise mechanisms of action.

## Figures and Tables

**Figure 1 metabolites-14-00563-f001:**
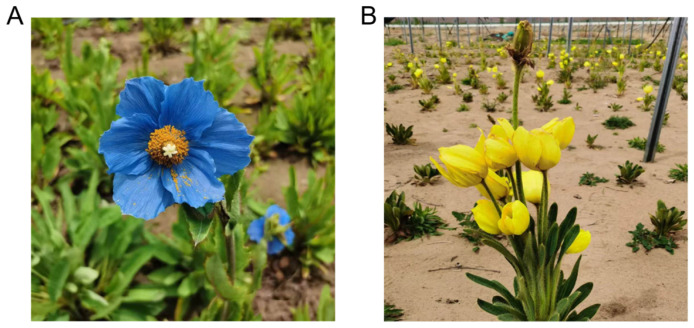
Photographs of two *Meconopsis* species. (**A**) *Meconopsis betonicifolia* (blue-flowered); (**B**) *Meconopsis integrifolia* (yellow-flowered).

**Figure 2 metabolites-14-00563-f002:**
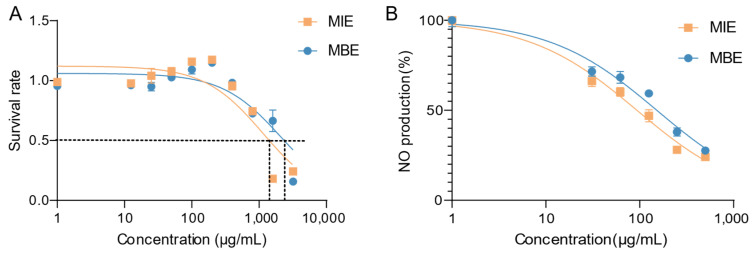
Effects of MIE (yellow) and MBE (blue) extracts on RAW264.7 cell viability and nitric oxide (NO) production under inflammatory conditions. (**A**) Cytotoxicity profiles: cell survival rate. (**B**) NO inhibition rate in LPS-stimulated cells.

**Figure 3 metabolites-14-00563-f003:**
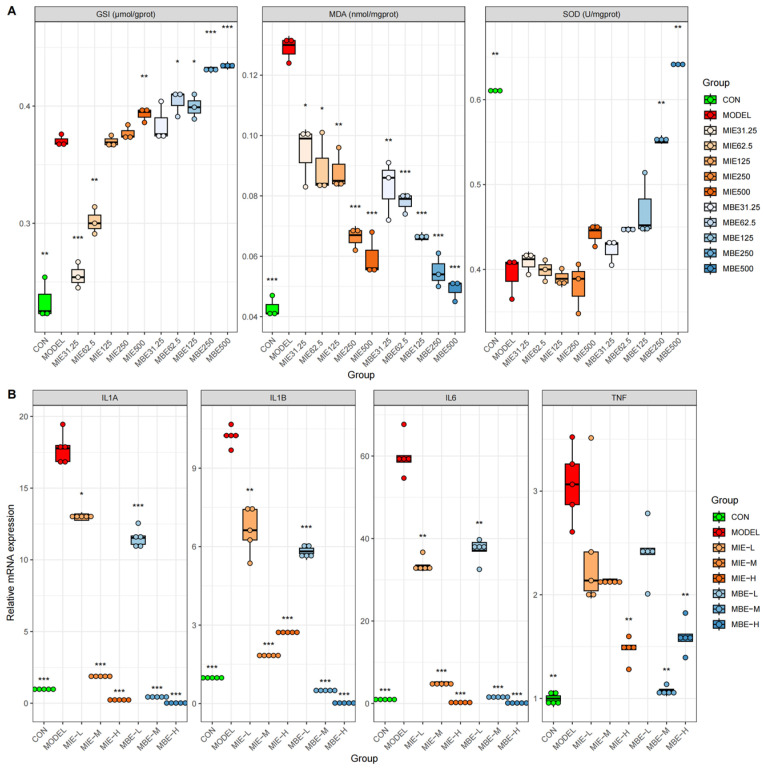
Antioxidant and anti-inflammatory effects of MIE (yellow) and MBE (blue). (**A**) Impact on oxidative stress markers GSH, MDA, and SOD levels in cells. (**B**) Effects on inflammatory gene expression of IL-1A, IL-1B, IL-6, and TNF in cells. ** p* < 0.05, ** *p* < 0.01, *** *p* < 0.001, vs. the model group.

**Figure 4 metabolites-14-00563-f004:**
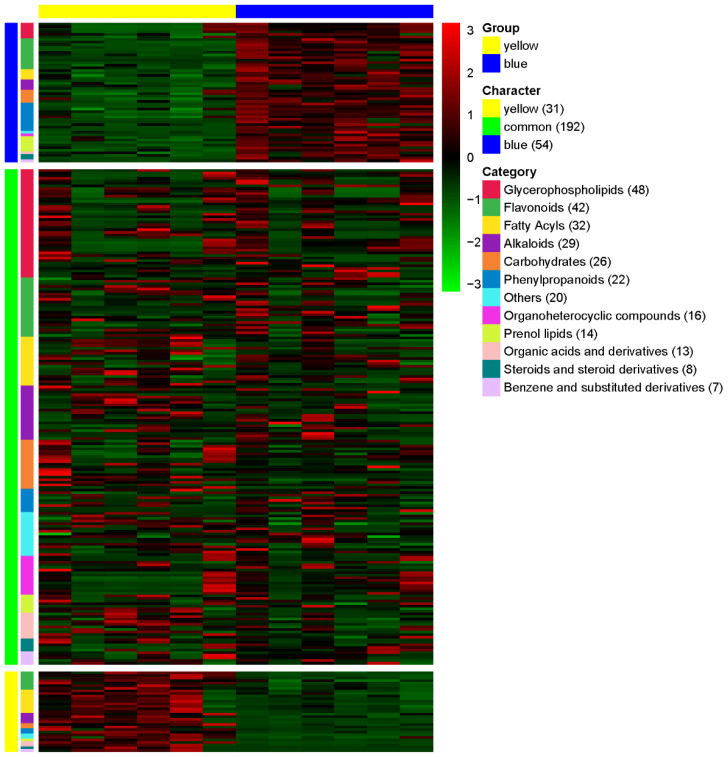
Comparative metabolomics analysis of extracts of yellow (MIE) and blue (MBE) *Meconopsis* varieties. The heatmap illustrates the relative abundance of compounds across different categories in the two *Meconopsis* species. Compounds are classified into 12 major categories and further grouped as yellow-specific, common to both species, or blue-specific. The color scale ranges from blue (lower abundance) to red (higher abundance).

**Figure 5 metabolites-14-00563-f005:**
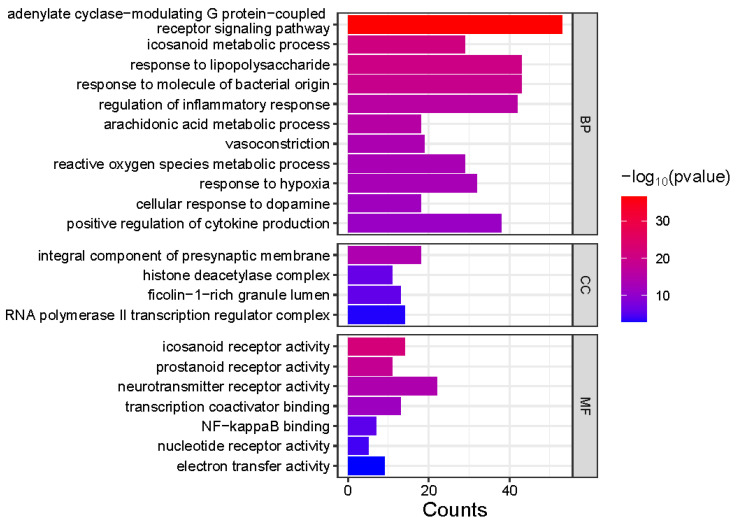
Gene Ontology (GO) enrichment analysis of common feature targets in *Meconopsis* species. GO enrichment analysis of 413 genes from two *Meconopsis* varieties. GO terms are categorized into Biological Process (BP), Cellular Component (CC), and Molecular Function (MF). The color intensity represents the −log10(*p*-value) of enrichment, while the size of each square indicates the count of genes associated with each GO term.

**Figure 6 metabolites-14-00563-f006:**
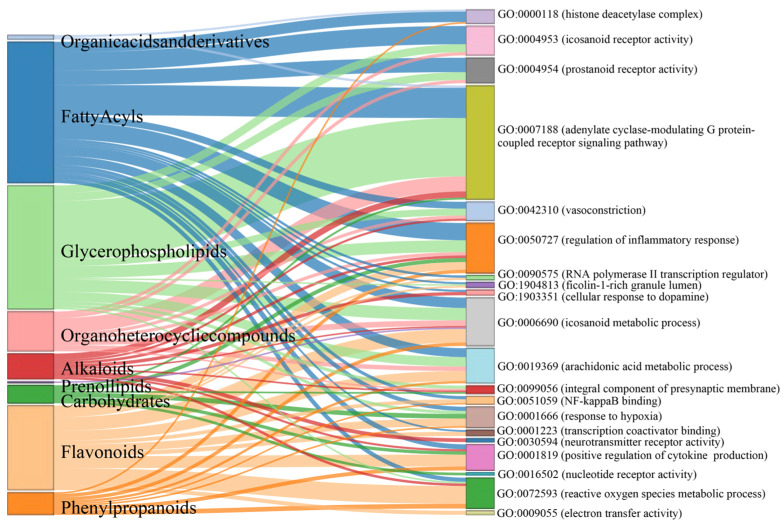
Sankey diagram illustrating the relationships between major compound categories (**left**) and their associated enriched GO terms (**right**) identified in the common feature targets of both yellow and blue *Meconopsis* species. The width of the flows represents the strength of the association between compound categories and GO terms.

**Figure 7 metabolites-14-00563-f007:**
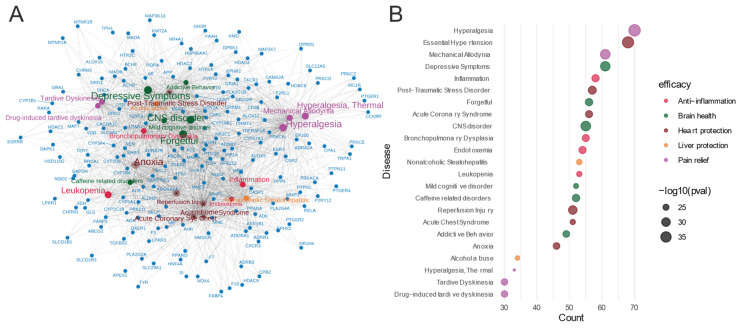
Gene-disease enrichment of common feature targets in *Meconopsis* species. (**A**) Gene–disease network. (**B**) Bubble plot of enriched diseases. Diseases were categorized by their corresponding therapeutic efficacy and represented by different colors. The size of the disease name and each bubble correlates with the adjusted *p*-value, where larger text sizes and bubbles indicate higher enrichment significance for the associated disease.

**Figure 8 metabolites-14-00563-f008:**
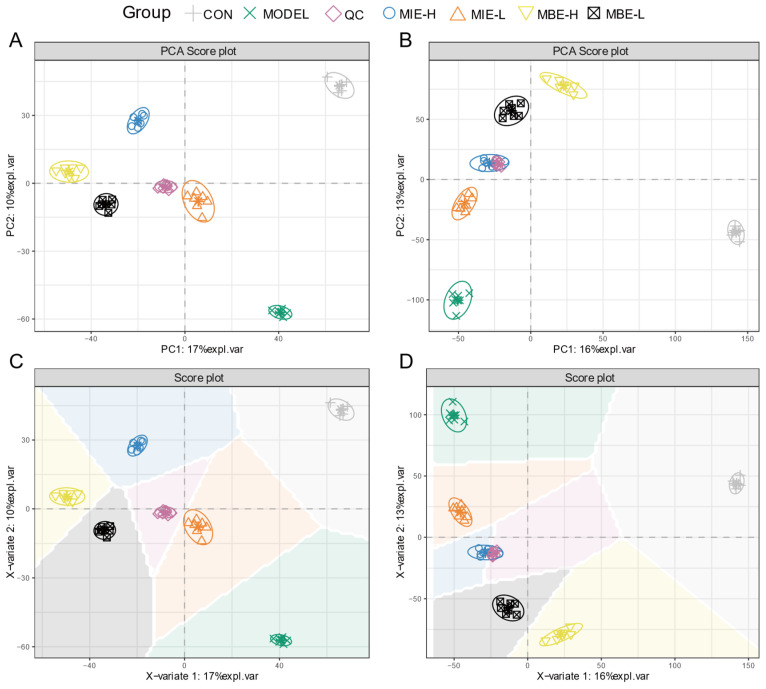
Metabolomics analysis of anti-inflammatory effects of *Meconopsis* varieties in LPS-stimulated RAW264.7 cells. (**A**) Positive ion mode PCA score plot. (**B**) Negative ion mode PCA score plot. (**C**) Positive ion mode PLS-DA score plot. (**D**) Negative ion mode PLS-DA score plot.

**Figure 9 metabolites-14-00563-f009:**
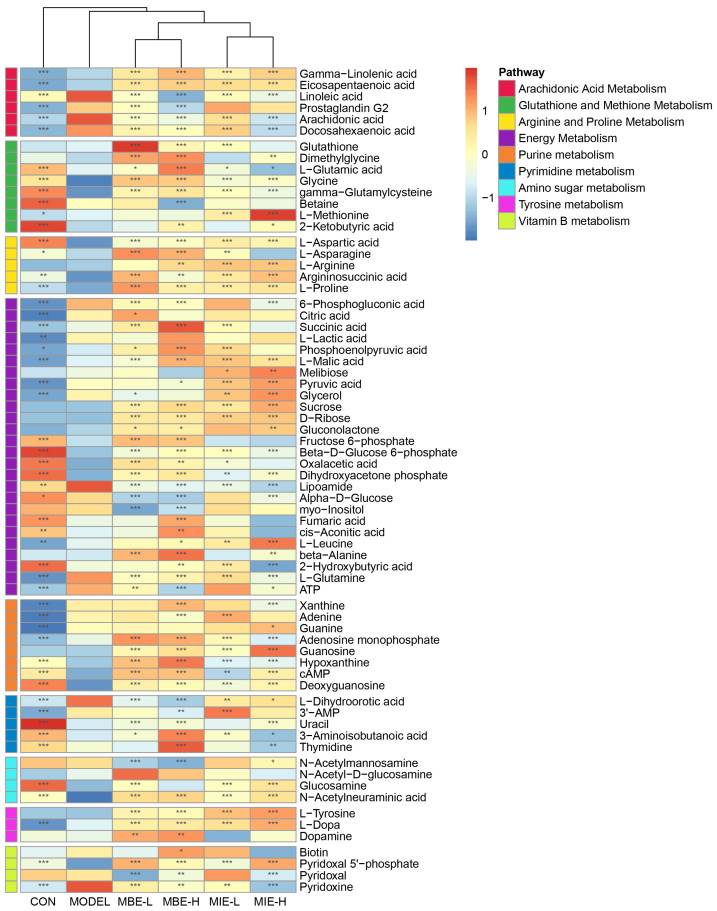
Heatmap of differentially abundant metabolites across key metabolic pathways in response to LPS stimulation and *Meconopsis* treatments. This heatmap illustrates the relative abundance of metabolites involved in nine key pathways: Arachidonic Acid Metabolism, Glutathione and Methionine Metabolism, Arginine and Proline Metabolism, Energy Metabolism, Purine Metabolism, Pyrimidine Metabolism, Amino Sugar Metabolism, Tyrosine Metabolism, and Vitamin B Metabolism. The color scale ranges from blue (lower abundance) to red (higher abundance). ** p* < 0.05, ** *p* < 0.01, *** *p* < 0.001, vs. the model group.

**Figure 10 metabolites-14-00563-f010:**
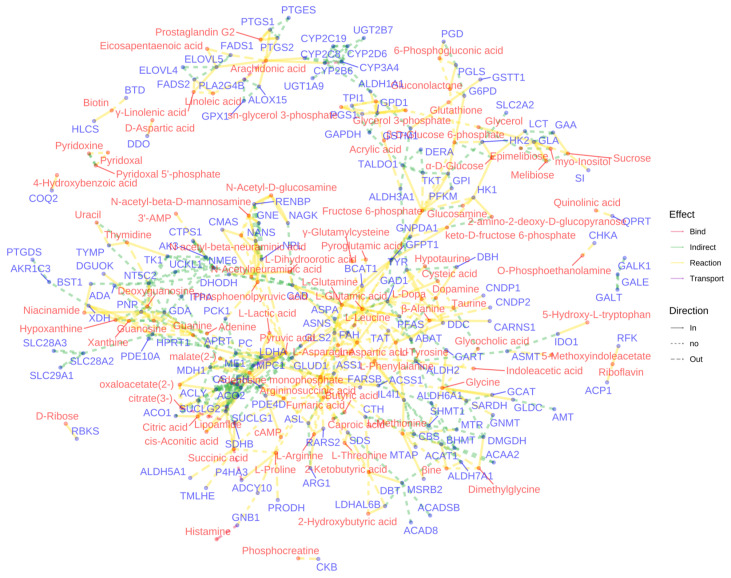
Network visualization of metabolite–gene interactions in enriched pathways. This figure presents a comprehensive network analysis of the interactions between differentially abundant metabolites and their corresponding genes within the enriched pathways. Nodes represent differential metabolites and their associated genes, and edges connecting these nodes signify the biochemical reactions or regulatory interactions that occur between them.

**Table 1 metabolites-14-00563-t001:** Primer sequences for real-time quantitative PCR in mice.

Gene	Forward Primer (5′-3′)	Reverse Primer (3′-5′)
IL-1B	GAAATGCCACCTTTTGACAGTG	TGGATGCTCTCATCAGGACAG
IL-1A	TCTCAGATTCACAACTGTTCGTG	AGAAAATGAGGTCGGTCTCACTA
TNF	GTAGCCCACGTCGTAGCAAA	ACAAGGTACAACCCATCGGC
IL-6	TGGAGTACCATAGCTACCTGGA	TGGAAATTGGGGTAGGAAGGAC
β-actin	GATATCGCTGCGCTGGTCG	CATTCCCACCATCACACCCT

## Data Availability

The original contributions presented in the study are included in the article/Appendix A; further inquiries can be directed to the corresponding author.

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
