# Peer review of "Comparative Evaluation of the Chemical Components and Anti-Inflammatory Potential of Yellow- and Blue-Flowered Meconopsis Species: M. integrifolia and M. betonicifolia"

_metabolites, 2024, doi:10.3390/metabo14100563_

Round 1
Reviewer 1 Report
Comments and Suggestions for Authors
The authors in this study compared the anti-inflammatory potential of yellow-flowered Meconopsis (M. integrifolia) with the scarce blue-flowered Meconopsis (M. betonicifolia). Both species demonstrated dose-dependent anti-oxidative and anti-inflammatory effects, with yellow Meconopsis showing superior efficacy. The research confirmed that yellow-flowered Meconopsis could be a viable alternative for anti-inflammatory use, supporting its clinical relevance and potential for sustainable utilization and biodiversity conservation. The manuscript is well-written and well-organized, has a logical flow of experiments, and will be of interest to the science community.
The following suggestions are recommended for improvement of the manuscript:
1. I recommend increasing the font size of the text in Figure 6, which is hard to read. It is better to have a clear-resolution figure.
2. In Figure 9, a few texts are hard to read. Improving the resolution will help the reader.
Author Response
Dear Reviewer,
Thank you for your insightful comments and suggestions, which are invaluable for enhancing the quality of our manuscript. In response to your feedback, we have made the following revisions to address the concerns raised:
Comments 1: I recommend increasing the font size of the text in Figure 6, which is hard to read. It is better to have a clear-resolution figure.
Response 1: Regarding the issue of text size in Figure 6, we have divided Figure 6 into parts A and B, reduced the color contrast of the lines connecting the data points, and increased the font size and weight of the text, thereby ensuring greater clarity and emphasis on the textual information.
Comments 2:In Figure 9, a few texts are hard to read. Improving the resolution will help the reader.
Response 2: Concerning the readability of certain texts in Figure 9, we have increased the resolution of the figure and bolded and enlarged the relevant fonts to ensure that readers can clearly discern the information presented in the image.
We sincerely appreciate your constructive comments, which have guided us in refining the manuscript's quality. We remain committed to continuous improvement and look forward to making further contributions to the academic community through our ongoing research endeavors.
Thank you again for your valuable feedback.
Sincerely,
Junsong Wang
Center of Molecular Metabolism, Nanjing University of Science and Technology, Nanjing 210094, China
E-mail: wangjunsong@njust.edu.cn
Reviewer 2 Report
Comments and Suggestions for Authors
This exhaustive study evaluating and comparing the anti-inflammatory potential of 18 yellow- and blue-flowered Meconopsis species, particularly M. integrifolia and M. betonicifolia, is timely and relevant given the overexploitation of well-known medicinal herbs.
While the topic is novel and fills a gap in the field by expanding anti-inflammatory options for M. betonicifolia, the paper could benefit from reducing its length, particularly by halving the introduction. Merging the results and discussion sections would enhance comprehensibility.
The statistical analysis and metabolomic study require further clarification to ensure reader understanding.
However, the conclusions are consistent with the presented evidence and arguments, effectively addressing the main research question.
The references are appropriate, and the tables and figures effectively convey the key findings.
Comments on the Quality of English Languageit can be improved.
Author Response
Dear Reviewer,
We sincerely appreciate your thorough review and insightful comments, which have significantly contributed to enhancing the quality and clarity of our manuscript. In response to your feedback, we have implemented the following revisions:
Comments 1: While the topic is novel and fills a gap in the field by expanding anti-inflammatory options for M. betonicifolia, the paper could benefit from reducing its length, particularly by halving the introduction. Merging the results and discussion sections would enhance comprehensibility.
Response 1: As suggested, we have condensed and streamlined the Introduction, Discussion and Conclusion sections, eliminating redundancies and streamlining the language to enhance comprehensibility while preserving the core arguments and conclusions supported by the presented evidence.
Comments 2: The statistical analysis and metabolomic study require further clarification to ensure reader understanding.
Response 2: In the Methods section, we have supplemented the manuscript with a new subsection (2.8) dedicated to statistical analysis, ensuring a clear understanding of the analytical approaches employed in our study.
We appreciate your acknowledgment of the novelty and relevance of our topic, as well as the effectiveness of our tables and figures in conveying the key findings. Your insightful feedback has undoubtedly strengthened the quality and clarity of our manuscript.
Thank you again for your valuable contributions to improving our work. We are grateful for the opportunity to address your comments and enhance the overall presentation and impact of our research.
Sincerely,
Junsong Wang
Center of Molecular Metabolism, Nanjing University of Science and Technology, Nanjing 210094, China
E-mail: wangjunsong@njust.edu.cn
Reviewer 3 Report
Comments and Suggestions for Authors
In this manuscript, the authors adopted a metabolomic approach to comparatively investigate the phytochemistry and the anti-inflammatory potential of yellow- and blue-flowered Meconopsis. Overall, the authors have done sufficient work and presented adequate interesting results to warrant publication. Nonetheless, some minor amendments are still required, which I have listed below.
1. ABSTRACT:
· Lines 15-16: The abstract can begin with a more convincing and accurate tone by indicating briefly the key diseases that Meconopsis is used to treat, instead of just stating “various diseases”. Importantly, since the study dealt with anti-inflammatory effects, the authors could consider stating any inflammation-associated diseases that the species has been used to treat.
· Line 20: The abbreviation “LC-MS” should be introduced in full when first mentioned.
· Line 31-32: “… highlighting clinical relevance and potential for sustainable utilization and biodiversity conservation.” – Since the study did not involve any clinical work or analysis of sustainability parameters, writing this in the conclusion may seem not very convincing.
2. INTRODUCTION:
· This is optional but I think it would be good to have - if the authors could include photographs showing the two species in the paper.
· Some statements, e.g., lines 66-70, seem inappropriate here and should be moved to RESULTS/DISCUSSION.
· In the last two paragraphs, some statements using “we will ...” and “we aim to…” should be revised since they are referring to a study that has already been conducted.
3. MATERIALS AND METHODS:
· Line 117: “Meconopsis betonicifolia and Meconopsis integrifolia” should be revised to “M. betonicifolia and M. integrifolia” as per convention of simplifying the genus after the first time it is mentioned.
· Lines 122-130: The authors have described how they did their solvent extraction using 70% ethanol. Was this done following a certain reference? If so, it should be cited. Importantly, is there any specific rationale of using 70% ethanol for this study? Is it because the 70% ethanol extraction is believed to best represent/simulate the way key active compounds can be extracted when the plants are used in traditional Tibetan medicine?
· Please make sure all abbreviations are introduced when first mentioned, e,g. see “MTT”, on line 145. Also, a reference should be cited for the method of the MTT assay.
· Please recheck to make sure that all necessary references have been cited to support the M&M. For example, see line 217 “… was performed using R package mixOmics.” The authors of the tool request that users cite their paper. Please see http://mixomics.org/cite-us/.
· When using online tools/databases, it would be conventional to indicate the access dates too. Please recheck whether this is appropriate for the webpages indicated in lines 238-256.
· Since statistical analysis has been performed, the authors could consider adding in a short second about statistical analysis, providing information such as number of replication, error bars, statistical software used, etc.
4. RESULTS:
· Please move each figure to the nearest position after it is first mentioned in the text. For example, please move Figures 1, 2, 4, 5, 7, and 8.
· Line 265: “… at therapeutically relevant concentrations.” – Is there any reference to support the statement that the concentrations tested were “therapeutically relevant”?
· Lines 269-272: Please recheck whether the description of results here is accurate. Based on Figure 1B, the NO production levels for both MBE and MIE look very similar. Have the authors performed any statistical analysis to confirm that the two datasets as well as the two IC50 values derived from them are indeed statistically significant? Although the authors wrote “MIE showed superior potency in reducing NO levels compared to MBE”, if the two datasets indeed have any statistically significant difference, MIE (shown to be allowing higher NO production in Fig 1B) looks weaker than MBE in suppressing NO production, thus suggesting weaker anti-inflammatory potential. Please recheck this part.
· Figure 1 caption: Please make it more concise and remove repetitive information, e.g., “Dose-dependent inhibition of NO production: NO inhibition rate in LPS-stimulated cells”.
· Lines 303-304: “IL-1B and IL-6 expression exhibited the most pronounced dose-dependent decrease for both species” – The statement seems untrue for IL-1B. Please see Figure 2, where the level for MIE-H is higher than that of MIE-M in the IL-1B chart. Please recheck the statement.
· Lines 304 & 313: The yellow variety is indicated as MBE, which is the same as the blue variety - is this a typo error? Please recheck the whole text for this kind of error.
· In lines 307 and 315, the authors have described some trends and differences as “subtle” and “marginally superior”. However, observations like these can be subjective and may not accurately reflect any true statistical significance of the findings. I would strongly recommend that the authors run statistical tests to confirm whether the differences described throughout the whole paper are indeed statistically significant. Any differences that are not statistically significant may not be as meaningful as implied. Statistical analysis is crucial to ensure the reliability/validity of the conclusions made in this study.
· Figure 2 – The charts need to be improved in a few ways. For example, instead of “value” (see y axis), it would be more meaningful/informative to indicate the units of the parameters measured. Furthermore, since the caption indicates just two colors (yellow and blue), it would be less confusing to avoid using different tones of the yellow and blue in the symbols. In fact, since the information is already clearly provided on the x axis, the different tones of yellow and blue seem redundant.
· Figure 5 – I would suggest that the texts superimposing on diagram be moved to the white areas on both sides of the diagram – to make it easier to read.
· Figure 6 – “A” and “B” (already mentioned in the caption) should also be indicated right next to be illustrations/chart. Overall, they should be enlarged to make the font size larger and easier to read.
5. DISCUSSION:
· Some parts look like RESULTS, e.g., lines 454-467, and can be made more concise.
· Lines 553-558: Has this been found for any other medicinal plants as well? Or something unique to the two species investigated in this study?
· Line 629-631: Reference should be cited to support this statement about dopamine.
· Lines 632-641 can be merged with the CONCLUSION, consolidating the information and excluding any repetitive information.
6. CONCLUSION:
· The section should be made more concise and straight-to-be-point, eliminating interpretation/speculative statements, which are more appropriate to DISCUSSION (e.g., see lines 652-654).
7. The authors reported 70% chemical similarity between the yellow- and blue-flowered Meconopsis species. Did they have any information regarding the relative abundance of key active compounds between the two species? Would it be a factor that could influence the relative potency between the two species?
Author Response
Dear Reviewer,
We would like to express our sincere gratitude for your thorough review and invaluable suggestions, which have greatly helped us to further improve the quality of our manuscript. We have carefully addressed each of your comments as detailed below:
- Abstract:
Comments:
- Lines 15-16: The abstract can begin with a more convincing and accurate tone by indicating briefly the key diseases that Meconopsis is used to treat, instead of just stating “various diseases”. Importantly, since the study dealt with anti-inflammatory effects, the authors could consider stating any inflammation-associated diseases that the species has been used to treat.
- Line 20: The abbreviation “LC-MS” should be introduced in full when first mentioned.
- Line 31-32: “… highlighting clinical relevance and potential for sustainable utilization and biodiversity conservation.” – Since the study did not involve any clinical work or analysis of sustainability parameters, writing this in the conclusion may seem not very convincing.
Response:
- Lines 15-16, we have revised "various diseases" to "various inflammatory and pain-related conditions" and provided examples of inflammatory diseases such as pneumonia and hepatitis that are treated with Meconopsis in the Introduction (lines 49-51).
- Line20, we have added the full term "Liquid chromatography-mass spectrometry (LC-MS) techniques".
- Lines 33-37, as per your recommendation, we have modified the conclusion to focus on the potential of the more abundant yellow-flowered Meconopsis to serve as a substitute for the scarce blue-flowered resources, providing experimental evidence for future clinical application. Certainly, since the text does not involve the clinical aspect, we have removed the ambiguous part “clinical relevance and”.
- Introduction:
Comments:
- This is optional but I think it would be good to have - if the authors could include photographs showing the two species in the paper.
- Some statements, e.g., lines 66-70, seem inappropriate here and should be moved to RESULTS/DISCUSSION.
- In the last two paragraphs, some statements using “we will ...” and “we aim to…” should be revised since they are referring to a study that has already been conducted.
Response:
- Following your suggestion, we have added photographs of the blue-flowered and yellow-flowered Meconopsis species (Figure 1).
- We have streamlined the Introduction to focus on the scientific question of whether the yellow-flowered variety can serve as a viable alternative to the blue-flowered Meconopsis due to resource scarcity, and how we have utilized modern techniques to explore the chemical and pharmacological similarities between the two.
- Thank you for your insightful comments regarding Lines 66-70“Our preliminary LC-MS analysis revealed a high degree of chemical similarity (over 70%) between the two species, providing a crucial foundation for exploring the medicinal potential of the yellow-flowered Meconopsis as an alternative to the blue variety”. We have carefully considered your feedback and have made significant revisions to consolidate the relevant content into the Results, Discussion, and Conclusion sections.
- Materials and Methods:
Comments:
- Line 117: “Meconopsis betonicifolia and Meconopsis integrifolia” should be revised to “M. betonicifolia and M. integrifolia” as per convention of simplifying the genus after the first time it is mentioned.
- Lines 122-130: The authors have described how they did their solvent extraction using 70% ethanol. Was this done following a certain reference? If so, it should be cited. Importantly, is there any specific rationale of using 70% ethanol for this study? Is it because the 70% ethanol extraction is believed to best represent/simulate the way key active compounds can be extracted when the plants are used in traditional Tibetan medicine?
- Please make sure all abbreviations are introduced when first mentioned, e,g. see “MTT”, on line 145. Also, a reference should be cited for the method of the MTT assay.
- Please recheck to make sure that all necessary references have been cited to support the M&M. For example, see line 217 “… was performed using R package mixOmics.” The authors of the tool request that users cite their paper. Please see http://mixomics.org/cite-us/.
- When using online tools/databases, it would be conventional to indicate the access dates too. Please recheck whether this is appropriate for the webpages indicated Lines 238-256.
- Since statistical analysis has been performed, the authors could consider adding in a short second about statistical analysis, providing information such as number of replication, error bars, statistical software used, etc.
Response:
- Line100, we have used the established abbreviation as it has been previously introduced.
- Lines 105-109, we have provided the reference supporting the use of 70% ethanol extraction.
- Lines 129-131, we have spelled out "Methyl thiazolyl tetrazolium (MTT)" and cited the latest reference for the detection kit.
- Line205, we have cited the appropriate reference for the mixOmics software used for PCA and PLS-DA analyses.
- Lines 226-239, we have verified the correspondence between the cited websites and online databases.
-
We have supplemented the manuscript with a new subsection (2.8) dedicated to statistical analysis, ensuring a clear understanding of the analytical approaches employed in our study.
- Results:
Comments:
- Line 265: “… at therapeutically relevant concentrations.” – Is there any reference to support the statement that the concentrations tested were “therapeutically relevant”?
- Lines 269-272: Please recheck whether the description of results here is accurate. Based on Figure 1B, the NO production levels for both MBE and MIE look very similar. Have the authors performed any statistical analysis to confirm that the two datasets as well as the two IC50 values derived from them are indeed statistically significant? Although the authors wrote “MIE showed superior potency in reducing NO levels compared to MBE”, if the two datasets indeed have any statistically significant difference, MIE (shown to be allowing higher NO production in Fig 1B) looks weaker than MBE in suppressing NO production, thus suggesting weaker anti-inflammatory potential. Please recheck this part.
- Figure 1 caption: Please make it more concise and remove repetitive information, e.g., “Dose-dependent inhibition of NO production: NO inhibition rate in LPS-stimulated cells”.
- Lines 303-304: “IL-1B and IL-6 expression exhibited the most pronounced dose-dependent decrease for both species” – The statement seems untrue for IL-1B. Please see Figure 2, where the level for MIE-H is higher than that of MIE-M in the IL-1B chart. Please recheck the statement.
- Lines 304 & 313: The yellow variety is indicated as MBE, which is the same as the blue variety - is this a typo error? Please recheck the whole text for this kind of error.
- Lines 307 and 315, the authors have described some trends and differences as “subtle” and “marginally superior”. However, observations like these can be subjective and may not accurately reflect any true statistical significance of the findings. I would strongly recommend that the authors run statistical tests to confirm whether the differences described throughout the whole paper are indeed statistically significant. Any differences that are not statistically significant may not be as meaningful as implied. Statistical analysis is crucial to ensure the reliability/validity of the conclusions made in this study.
- Figure 2 – The charts need to be improved in a few ways. For example, instead of “value” (see y axis), it would be more meaningful/informative to indicate the units of the parameters measured. Furthermore, since the caption indicates just two colors (yellow and blue), it would be less confusing to avoid using different tones of the yellow and blue in the symbols. In fact, since the information is already clearly provided on the x axis, the different tones of yellow and blue seem redundant.
- Figure 5 – I would suggest that the texts superimposing on diagram be moved to the white areas on both sides of the diagram – to make it easier to read.
- Figure 6 – “A” and “B” (already mentioned in the caption) should also be indicated right next to be illustrations/chart. Overall, they should be enlarged to make the font size larger and easier to read.
Response:
- Lines 259-261, as suggested, we have emphasized that the cell viability remained above 90% at the 400 μg/mL concentration, highlighting the safety of this range.
- We have carefully checked and corrected the descriptions of the blue-flowered and yellow-flowered varieties, clarifying Lines 270-272 that the yellow-flowered Meconopsis(MIE) exhibited more potent inhibition of nitric oxide.
- As recommended, we have removed the redundant content from the Figure 1 (original Figure 2) caption and revised it to "NO inhibition rate in LPS-stimulated cells".
- Lines 310-314, we appreciate your meticulous feedback. We have corrected the erroneous description of MIE and clarified that only IL-6 showed a dose-dependent downregulation, while MBE was more effective in inhibiting IL-6. We have added statistical significance annotations to Figure 3 (original Figure 2) and removed the unnecessary information from the y-axis.
- Regarding your comments on the "subtle differences" and "nuanced differences" expressions, we have removed the paragraph describing the nuanced differences between the two Meconopsis species, as the details provided were either inaccurate or overly specific.
- In Figure 7 (original Figure 6), we have labeled the positions of A and B and increased the font size for better readability. Similarly, we have bolded and enlarged the fonts in Figure 6 (original Figure 5).
- Discussion and Conclusion:
Comments:
- Some parts look like RESULTS, e.g., lines 454-467, and can be made more concise.
- Lines 553-558: Has this been found for any other medicinal plants as well? Or something unique to the two species investigated in this study?
- Lines 632-641 can be merged with the CONCLUSION, consolidating the information and excluding any repetitive information.
- Conclusion: The section should be made more concise and straight-to-be-point, eliminating interpretation/speculative statements, which are more appropriate to DISCUSSION (e.g., see lines 652-654).
Response:
- We have streamlined these sections, removing repetitive and overly speculative statements.
- Regarding your question on lines 553-558, our inferences were based on the observed changes in metabolites and associated biochemical pathways, considering their biological significance, and not specific to the two Meconopsis species. We have revised this part accordingly.
- We have merged and eliminated the redundant content between the Discussion and Conclusion sections.
- Other comments:
Comments:
- The authors reported 70% chemical similarity between the yellow- and blue-flowered Meconopsisspecies. Did they have any information regarding the relative abundance of key active compounds between the two species? Would it be a factor that could influence the relative potency between the two species?
Response:
Regarding your seventh point on the relative abundance of the main active compounds in the two Meconopsis species, we have described in the Results section 3.3 that "The results demonstrated a high degree of similarity in chemical composition between the two-color varieties, with over 70% of the detected compounds present in both." We have also provided the information on the detected compounds in Supplementary Table S1.
Once again, we sincerely appreciate your valuable feedback, which has significantly enhanced the quality of our manuscript. We will continue our efforts to make further contributions to the academic community.
Sincerely,
Junsong Wang
Center of Molecular Metabolism, Nanjing University of Science and Technology, Nanjing 210094, China
E-mail: wangjunsong@njust.edu.cn